# Facile Formation of Anatase/Rutile TiO_2_ Nanocomposites with Enhanced Photocatalytic Activity

**DOI:** 10.3390/molecules24162996

**Published:** 2019-08-19

**Authors:** Jing He, Yi-en Du, Yang Bai, Jing An, Xuemei Cai, Yongqiang Chen, Pengfei Wang, Xiaojing Yang, Qi Feng

**Affiliations:** 1School of Chemistry & Chemical Engineering, Jinzhong University, Jinzhong 030619, China; 2State Key Laboratory of Coal Conversion, Institute of Coal Chemistry, Chinese Academy of Sciences, Taiyuan 030001, China; 3Beijing Key Laboratory of Energy Conversion and Storage Materials, College of Chemistry, Beijing Normal University, Beijing 100875, China; 4Department of Advanced Materials Science, Faculty of Engineering, Kagawa University, 2217-20 Hayashi-cho, Takamatsu-shi 761-0396, Japan

**Keywords:** anatase/rutile nanocomposites, {101} facets, {110} facets, photocatalytic activity

## Abstract

Anatase/rutile mixed-phase TiO_2_ nanoparticles were synthesized through a simple sol-gel route with further calcination using inexpensive titanium tetrachloride as a titanium source, which effectively reduces the production cost. The structural and optical properties of the prepared materials were characterized by X-ray diffraction (XRD), transmission electron microscopy (TEM), and UV-vis adsorption. The specific surface area was also analyzed by Brunauer–Emmett–Teller (BET) method. The anatase/rutile mixed-phase TiO_2_ nanocomposites containing of rod-like, cuboid, and some irregularly shaped anatase nanoparticles (exposed {101} facets) with sizes ranging from tens to more than 100 nanometers, and rod-like rutile nanoparticles (exposed {110} facets) with sizes ranging from tens to more than 100 nanometers. The photocatalytic activities of the obtained anatase/rutile mixed-phase TiO_2_ nanoparticles were investigated and compared by evaluating the degradation of hazardous dye methylene blue (MB) under ultraviolet light illumination. Compared to the commercial Degussa P25-TiO_2_, the mixed-phase TiO_2_ nanocomposites show better photocatalytic activity, which can be attributed to the optimal anatase to rutile ratio and the specific exposed crystal surface on the surface. The anatase/rutile TiO_2_ nanocomposites obtained at pH 1.0 (pH1.0-TiO_2_) show the best photocatalytic activity, which can be attributed to the optimal heterojunction structure, the smaller average particle size, and the presence of a specific exposed crystal surface. The enhanced photocatalytic activity makes the prepared anatase/rutile TiO_2_ photocatalysts a potential candidate in the removal of the organic dyes from colored wastewater.

## 1. Introduction

Water pollution resulting from textile dyes and other industrial dyestuffs has become an overwhelming problem worldwide [1,2]. The colored wastewater discharged into the environment during the dying process is considered to the main source of environmental hazards, such as nonaesthetic pollution and eutrophication. In addition, the dangerous byproducts formed as a result of oxidation, hydrolysis, or other chemical reactions taking place in the wastewater are considered to endanger human health [1,2]. Hence, removing above the organic dyes from the colored wastewater has attracted widely attention and deeply research. In the past decades, titanium dioxide (TiO_2_) has been extensively used as the photocatalyst for degradation of the organic contaminants in the environment under ultraviolet (UV) or visible light irradiation, because of its high stability, strong redox ability, nontoxicity, good corrosion resistance, and low cost [3,4]. TiO_2_ has three kinds of phases, anatase (tetragonal, space group *I*4_1_/*amd*), rutile (tetragonal, space group *P*42_1_/*mmm*), and brookite (orthorhombic, space group *Pbca*), among which anatase is mostly used as photocatalysts, since it is traditionally considered that the anatase phase has lower rates of recombination and smaller grain size than the other phases, which may be favorable and preferable to achieve better photocatalytic efficiency [1,5]. However, the band gap of anatase is about 3.2 eV, which can only be excited by photons with wavelengths below 387 nm, i.e., anatase can show a photocatalytic activity only under UV light irradiation, while UV light accounts for only a small fraction (~5%) of the solar energy [1]. Therefore, various methods have been applied to modify anatase to make better use of solar energy, such as doping of the metal ions and nonmetal ions, modification of surface morphology, design of TiO_2_ heterostructures, controlling exposed facetsof TiO_2_ crystals, and so on [6]. It is reported that the combination of the anatase and rutile phases of TiO_2_ exhibits higher photocatalytic performance in the decomposition ofvarious organic pollutants than that of pure anatase or rutile phase due to the transfer of electrons from rutile to anatase TiO_2_ during photo-excitation, which inhibits the charge recombination of anatase, leading to more efficient separation of the photogenerated electron–hole pairs and greater photocatalytic activity [3,6,7]. Among the reported TiO_2_ materials, the commercial Degussa P25-TiO_2_ (87% anatase and 13% rutile phase) is often used as a benchmark model photocatalyst because of its superior photocatalytic activity [8]. Todate, various synthetic methods have been used to synthesize the mixed-phase of rutile and anatase. For example, Liu et al. prepared a heterogeneous anatase/rutile nanostructure by the layer-by-layer assembly technique, which exhibited better photocatalytic activity for decomposing gaseous acetadehyde than the original anatase or the rutile nanomaterials [9]. Kawahara et al. prepared anatase/rutile coupled particles by a dissolution-reprecipitation method [10]. Ohno et al. prepared an anatase/rutile mixture by simply mixing anatase and rutile TiO_2_ particles in water at different ratios or calcining pure anatase TiO_2_ powers at different temperatures [11]. Among them, the hydrothermal/solvothermal method and the sol-gel method are the two most commonly used methods for synthesizing precise and tunable morphology of mixed-phase TiO_2_ nanoparticles, due to the advantages of moderate reaction conditions and controllable reaction process, etc. [12,13]. Cao et al. reported the fabrication of a new type of heterostructure by coupling Sn-doped rutile TiO_2_ and N-doped anatase TiO_2_ using a sol-gel method, which showed a higher photocatalytic activity than the individual film of Sn-doped rutile TiO_2_ and N-doped anatase TiO_2_ under both ultraviolet and visible light irradiation [13]; Liu et al. reported a synthesis of anatase/rutile mixed-phase TiO_2_ at relatively low temperature by using a one-step nonaqueous route, which exhibited excellent photocatalytic activity for the degradation of Rhodamine B [1]. Recently, Honget al. prepared the Fe_3_O_4_/TiO_2_ nanocomposites by using ilmenite as a raw material under solvothermal conditions, which showed higher photocatalytic activity conversion of Rhodamine B into degraded or mineralized products than the best commercially available P25-TiO_2_ nanoparticles [14]. Peng et al. prepared crystalline anatase/rutile mixed phase Sm-C-TiO_2_ catalysts by an ordinary sol-gel method, which showed better photocatalytic activity than undoped TiO_2_ and the commercial Degussa P25-TiO_2_ for the photocatalytic degradation of methylene blue [15]. Atitar et al. prepared the mesoporous anatase–rutile TiO_2_ mixtures through a simple one step sol-gel process at different temperatures, which showed higher activity for the decomposition of both imazapyr and phenol, compared to the nonporous P25-TiO_2_ [16].

Besides heterojunction structure, TiO_2_ with tailored facets for photocatalysis also has attracted considerable attention in recent years. In 2007, Wen et al. successfully obtained anatase TiO_2_ nanocrystals with exposed {010} facets and controllable morphologies by using the exfoliated layered titanate nanosheets as the precursor under hydrothermal conditions [17]. After that, Yang et al. reported that single crystals with a high percentage of exposed {001} facets were synthesized by using hydrofluoric acid as the morphology control agent [18]. Lee et al. reported that truncated octahedral anatase bipyramids with exposed {001} and {101} facets were synthesized by a vapor-solid reaction growth method [19]. Recently, {101}-oriented rutile TiO_2_ nanorods were prepared under hydrothermal conditions, which exhibited efficient solar water-splitting performance [20]. TiO_2_ nanosheets with dominant {001} facets were synthesized in the absence of any fluorine precursor or hazardous materialat room temperature [21]. The tetragonal-facets-rod anatase TiO_2_ dominated by {100} facets were synthesized by using titanate nanofibers derived from alkali treatment as precursor, which exhibited smart electrorheological behavior underexternal electric field [22]. We also synthesized various morphologies of anatase TiO_2_ nanocrystals with exposed special high energy facets [23,24,25].

In this work, anatase/rutile mixed-phase TiO_2_ with exposed {101} facets and {110} facets was synthesized through a simple sol-gel route with further calcination. This approach uses rich analytical ethanol and deionized water as the solvent, inexpensive titanium tetrachloride as the titanium source, and low-cost ammonium fluoride as the capping agent, which effectively decreases the production cost. The anatase/rutile mixed-phase TiO_2_ nanocomposites have a smaller nanocrystal size and a narrower grain distribution. The photocatalytic activity of the obtained anatase/rutile nanocomposites was evaluated using the degradation tests of methylene blue (MB) under UV light illumination. The obtained anatase/rutile mixed-phase TiO_2_ exhibited excellent photocatalytic activity because of the synergistic effect of the optimum anatase to rutile radio, and the specific exposed crystal surface, compared to the commercial Degussa P25-TiO_2_. The improvement in photocatalytic activity makes the prepared anatase/rutile nanocomposite a potential candidate in wastewater purification.

## 2. Results and Discussion

Figure 1 displays the XRD patterns of anatase/rutile TiO_2_ composites synthesized by the sol-gel method using titanium tetrachloride as the titanium source. It is worth noting that both the XRD patterns of pH0.5-TiO_2_ and pH1.0-TiO_2_ show the same diffraction peaks, which can be ascribed to the tetragonal anatase phase (JCPDs file no. 21-1272, space group*I*4_1_/*amd*) and the tetragonal rutile phase (JCPDs file no. 21-1276, space group*P*4_2_/*mnm*), and rutile TiO_2_ is the major phase because it is the most stable among the three polymorphs (anatase, brookite, and rutile) under strong acidic conditions [26]. As shown in Figure 1, the diffraction peaks at 2θ = 25.64°, 38.26°, 48.40°, 63.00°, 69.26°, and 75.16° are, respectively, assigned to the reflections of the (101), (004), (200), (204), (116), and (215) crystal planes of anatase TiO_2_; while the peaks located at 2θ = 27.74°, 36.42°, 41.58°, 44.40°, 54.64°, 56.56°, 63.09°, 64.20°, 69.26°, and 69.83° are indexed to the (110), (101), (111), (210), (211), (220), (002), (310), (301), and (112) diffraction peaks of rutile TiO_2_.The proportion of anatase/rutile TiO_2_ composite can be calculated by the following formulas (1) and (2) [27]:(1)$$W_{A}=\frac{0.886I_{A}}{0.886I_{A}+I_{R}}$$
(2)$$W_{R}=\frac{I_{R}}{0.886I_{A}+I_{R}}$$
where *W*_A_ and *W*_R_ represent the mass fraction of anatase TiO_2_ and rutile TiO_2_, respectively. *I*_A_ and *I*_R_ represent the integral intensity of diffraction peaks on crystal surface of anatase TiO_2_ (101) and rutile TiO_2_ (110), respectively. As shown in Figure 1a, when the pH is 0.5, the integral intensity of diffraction peak of anatase TiO_2_ (101) crystal surface and rutile TiO_2_ (110) crystal surface in the composite is 47.6% (*I*_A_) and 100.0% (*I*_R_), respectively. Therefore, the proportion of anatase TiO_2_ and rutile TiO_2_ in the composite is 29.66% and 70.34%, respectively. As shown in Figure 1b, when the pH is 1.0, the integral intensity of diffraction peak of anatase TiO_2_ (101) crystal surface and rutile TiO_2_ (110) crystal surface in the composite is 50.3% (*I*_A_) and 100.0% (*I*_R_), respectively. Therefore, the proportion of anatase TiO_2_ and rutile TiO_2_ in the composite is 30.83% and 69.17%, respectively. The average crystallite size of pH0.5-TiO_2_ and pH1.0-TiO_2_ are readily calculated from XRD patterns to be 27.2 nm and 25.7 nm, respectively. The patterns exhibited that with increasing pH, the intense anatase and rutile peaks become weak, which is due to the decrease in crystalline size, implying that increasing pH inhibits the formation of rutile TiO_2_ and enhances the formation of anatase TiO_2_ during the hydrothermal reaction process at 180 °C for 24 h.

The morphology of the TiO_2_ nanoparticles was characterized by TEM and HRTEM, as shown in Figure 2 and Figure 3. pH0.5-TiO_2_ is a mixture of anatase and rutile phase (see Figure 1a) and has a rod-like particle morphology with a size about 50–75 nm in length and 20–30 nm in width, a cuboid particle morphology with a size about 20–50 nm in length and 20–40 nm in width, and some irregularly shaped particles (Figure 2a,b). The lattice fringes with spacings of 3.47 Å (or 3.49 Å) can be assigned to {101} facets of the rod-like anatase particles, which are parallel to the lateral surface, indicating that the lateral surface exposed {101} facets of the rod-like particles (Figure 2c). In addition, rod-like rutile nanoparticles with exposed {110} facets (*d* = 3.26 Å) on the lateral surface were also observed (Figure 2c). The lattice fringes with spacings of 3.46 Å can be assigned to {101} facets of the approximately spherical anatase particle (Figure 2d). Figure 2e,f show the HRTEM images of the cuboid anatase TiO_2_ nanocrystals. The (101) atomic planes with a lattice distance of 3.52 Å can be clearly seen, indicating that the two square surfaces of top and bottom are the {101} facets.

pH1.0-TiO_2_ is also amixture of anatase and rutile phase (see Figure 1b) and also has a rod-like particle morphology with a size about 80–130 nm in length and 15–30 nm in width, a cuboid particle morphology with a size about 25–50 nm in length and 10–30 nm in width, and some irregularly shaped particles (Figure 3a). The lattice fringes with spacings of 3.21Å (or 3.27 Å) can be assigned to {110} facets of the rod-like rutile particles (or the approximated rhombic rutile particles), which are parallel to the lateral surface, indicating that the lateral surface exposed {110} facets of the rod-like rutile particles (or the approximated rhombic rutile particles) (Figure 3b,d). In addition, rod-like (or cuboid, irregular morphology) anatase nanoparticles with exposed {101} facets (*d* = 3.46~3.52 Å) on the lateral surface were also observed, indicating that it exposed the {101} facets on the lateral surfaces (Figure 3c–f).

Figure 4 shows the nanoparticle size distributions of pH0.5-TiO_2_ and pH1.0-TiO_2_ measured from enlarged photograph of TEM images. The measured average nanoparticle sizes are 32.85 nm and 27.10 nm for pH0.5-TiO_2_ and pH1.0-TiO_2_, respectively, which are close to values obtained from the XRD calculation. It is worth noting that the order of the average crystallite sizes measured from TEM images is consistent with that of average crystallite sizes calculated from XRD patterns.

The photocatalytic activity of the pH0.5-TiO_2_, pH1.0-TiO_2_, and P25-TiO_2_ samples was evaluated by observing the photodegradation of MB (2.92 × 10^−5^ mol/L) in aqueous solution under low-pressure mercury lamp. The photocatalytic MB is thought to be degraded by the following reactions [28,29]:

anatase TiO_2_/rutile TiO_2_ + *hv*→ anatase TiO_2_/rutile TiO_2_(eˉ + h^+^)

h^+^ + H_2_O → ·OH + H^+^

h^+^ + OHˉ→ ·OH 

eˉ + O_2_→ O_2_ˉ

O_2_ˉ + H_2_O→ ·OOH + OHˉ

·OOH + H_2_O→H_2_O_2_ + ·OH

eˉ + ·OOH + H^+^ →H_2_O_2_

H_2_O_2_ →2·OH

·OH + MB → peroxides or hydroxylatedintermediates →→ degraded or mineralized products.

The above reactions are chosen because both the electrons and holes eventually participate in the formation of ·OH radicals, which degrade the MB. Figure 5 shows the UV-vis spectral changes of MB aqueous solution as a function of UV irradiation time in the presence of pH0.5-TiO_2_, pH1.0-TiO_2_, P25-TiO_2_, and absence of any other catalysts. The corresponding time-dependent photodegradation profiles of MB are shown in Figure 6. Figure 5a–c shows that the maximal absorption spectra of MB at 664 nm blue-shifts by as much as 34 nm from 664 to 630 nm, indicating that the N-demethylation occurred during the course of the photodegradation of the MB aqueous solution [30]. Examination of the spectral variation of Figure 5a–c suggests that MB is demethylated in a stepwise manner, that is, methyl groups are removed one by one as confirmed by the gradual shift of maximal absorption peak towards the shorter wavelength. During the initial period of the photodegradation of MB, competitive reactions between demethylation and cleavage of the MB chromophore ring structure (phenothiazine or thionine) occur, with demethylation predominating. Irradiation by UV light for longer times leads to further decomposition of the demethylated MB intermediates as indicated by changes in peak intensity at 630 nm [28,30]. However, in the absence of photocatalysts, the absorption spectra of MB changes relatively little for every measurement under UV light irradiation with the extension of irradiation time (Figure 5d). It is clearly seen from Figure 6 that pH1.0-TiO_2_ shows the highest photocatalytic activity and pH0.5-TiO_2_ shows slightly lower photocatalytic activity than pH1.0-TiO_2_, while P25-TiO_2_ shows much lower photocatalytic activity than pH1.0-TiO_2_. Totals of 85.8%, 82.6%, and 79.4% MB were photodegraded for pH1.0-TiO_2_, pH0.5-TiO_2_, and P25-TiO_2_, respectively, under UV irradiation for 90 min. It also can be seen that the blank sample shows the lowest photocatalytic activity, displaying only 8.9% photodegradation of MB under UV irradiation for 90 min.

It is well known that the photocatalytic activity of a catalyst is determined by many factors, such as crystalline phase, degree of crystallinity, crystal size, specific surface area, crystal morphology, dopant, doping amount, and heterojunction structure [31,32,33,34]. The average crystal sizes calculated from enlarged photographs of the TEM images are 32.85, 27.10, and 27.8 nm [35] for pH0.5-TiO_2_, pH1.0-TiO_2_, and P25-TiO_2_, respectively. Due to the quantum size effect, smaller average crystal size provides more powerful redox ability, resulting in a reduction in the net electron–hole recombination, thereby increasing the photocatalytic activity [3]. It is well known that the specific surface area increases with the decrease of particle size. For the above samples, the specific surface areas of pH 0.5-TiO_2_, pH1.0-TiO_2_, and P25-TiO_2_ anatase/rutile TiO_2_ samples is 47.5, 53.2, and 52.5 m^2^/g [24], respectively. Generally speaking, the photocatalytic activity will increase with the increase of catalytic surface area because of the degradation occurring on the surface of the catalyst [29]. However, the specific surface areas of the above samples area round 50 m^2^/g (≤3.2 m^2^/g difference), and the experimental error for BET analysis is around ±10%, indicating that the specific surface area is not an important factor affecting the photocatalytic activity in the above catalytic reactions. Thus, the improvement in photocatalytic activity of pH0.5-TiO_2_ and pH1.0-TiO_2_ nanoparticles should be attributed to factors other than the specific surface area. It is well known that the photocatalysis of a crystal also depends on the surface structure and surface activity of the crystal facets [36]. Based on the above discussion, pH0.5-TiO_2_ and pH1.0-TiO_2_ are both mixtures of anatase and rutile phases, similar to P25-TiO_2_. The heterojunction structure of the mixed anatase/rutile TiO_2_ can mediate the migration of photogenerated holes across the interface into rutile TiO_2_ and reduce the recombination of the photogenerated electrons and holes [37]. However, the anatase and rutile content is about 30% and 70% for pH0.5-TiO_2_, 31% and 69% for pH1.0-TiO_2_, and 87% and 13% for P25-TiO_2_ [38], respectively. Comparison of the above results shows that although the P25-TiO_2_ has a greater anatase/rutile ratio (87%/13% = 6.39/1), the photodegradation efficiency is less than those of the pH0.5-TiO_2_ (30%/70% = 0.43/1) and pH1.0-TiO_2_ (31%/69% = 0.45/1) samples, indicating that the presence of an optimum anatase to rutile radio in the biphasic system is beneficial to increase the photocatalytic efficiency [39]. The photocatalytic activity of the anatase/rutile mixed-phase TiO_2_ nanocomposites first increases with increasing rutile nanocrystals content (P25-TiO_2_(13%) ˂ pH1.0-TiO_2_(69%)), and then decreases (pH0.5-TiO_2_(70%) ˂ pH1.0-TiO_2_ (69%)). As the rutile nanoparticle increases, photogenerated electron–hole pairs in the anatase phase can be separated by transferring the electrons to the rutile nanoparticle, and the holes remain in the anatase nanoparticle, which reduces the recombination rate of anatase, leading to the improvement of the photocatalytic efficiency. Since both electrons and holes participate in the degradation reactions, both anatase and rutile nanoparticles must have access to the MB solution [3,29]. With the increase of rutile nanoparticles, this access to the anatase phase is reduced and eventually blocked. The synergic effect of the optimal ratio produces the excellent photocatalytic activity. For pH0.5-TiO_2_ and pH1.0-TiO_2_, the content of anatase and rutile, the particle morphology (rod-like, cuboid, and irregular morphology), and the exposed facets ({110}/{101}) are similar, so the difference in photocatalytic activity can be attributed to the difference of the specific surface area, anatase to rutile radio, and the particle size. Furthermore, as photocatalysis is a surface phenomenon, it is closely related to the exposure of the crystal surface [40]. Therefore, the surface activity of the crystal facets is also a key factor. Ohno et al. suggested that the {001} facets of anatase particles provide the oxidation sites and the {101} facets work as the reduction sites, thus proposing an anisotropic behavior for the e^−^ and h^+^ on the external anatase crystal structure. For rutile particles, it is suggested that the oxidation site is mainly on the {011} facets with higher electronic energy levels and the reduction site is mainly on the {110} facets with lower electronic energy levels; the difference in the energy levels drives the electrons and holes to different crystal facets. The synergistic effect between the {011} and {110} facets accelerates the separation of electrons and holes and suppresses the recombination of photogenerated electrons and holes, resulting in improvement of the photocatalytic activity [41]. On the other hand, since the dye adsorption reaction occurs on the crystal surface, the adsorption behavior of the dye depends on the crystal facets of the TiO_2_ surface [42]. For anatase particles, the adsorption energy increases in the order of nonspecific surface < {101} facets < {010} facets < {001} facets < {110} facets < {111} facets [43]. For rutile particles, the adsorption energy increases in the order of nonspecific surface < {110} facets (0.544 J/m^2^) < {100} facets (0.824 J/m^2^) < {001} facets (1.398 J/m^2^) [44]. It has been reported that the strong interaction between the dye molecule and the anatase or rutile crystals facets can improve the charge transfer rate from the molecule to the adsorption TiO_2_ surface, resulting in the enhancement of the photocatalytic activity [45]. Based on the above discussion, the pH0.5-TiO_2_ and pH1.0-TiO_2_ samples with dominant exposed {110} and {101} facets exhibited the higher photocatalytic activity, compared with that of P25-TiO_2_ without an exposed specific surface. P25-TiO_2_ shows the lower photocatalytic activity, which can be attributed to the absence of optimal anatase to rutile ratio and specific exposed crystal surface [35].

## 3. Materials and Methods

### 3.1. Synthesis of Anatase/Rutile TiO_2_ Composite Nanoparticles

In synthesis of the nanoparticles, 50 mL titaniumtetrachloride (purity 99.0%, Tianjin Guangfu Fine Chemical Research Institute) was added dropwise into 150 mL analytical ethanol (purity 99.7%, Tianjin Kemio Chemical Reagent Co., LTD, Tianjin, China) under vigorous stirring. After 30 min, the above solution was added dropwise into 440 mL deionized water to obtain a colloidal solution under continuous stirring. After 24 h, 50 mL of the above colloidal solution was transferred into two Teflon autoclaves with a capacity of 80 mL and the pH was adjusted to 0.5 and 1.0, respectively. Then, 0.20 g of ammonium fluoride solid was added into the above Teflon autoclaves under vigorous stirring. After that, the temperature of the above autoclaves was raised to 180 °C, maintained for 24 h, and then cooled to room temperature. Two white colloidal precipitates were obtained at the bottom of the autoclaves. The white colloidal precipitates were isolated from the solution by filtering, and were washed with deionized water 4 or 5 times. After the white colloidal precipitates were dried overnight at 80 °C, two white gels were obtained. The white gels were grounded carefully and then annealing at 450 °C for 1 h in a high temperature box furnace.

### 3.2. CharacterizationMethods

The structures of crystalline samples were investigated by using a powder X-ray diffractometer (XRD, Rigaku MiniFlex II desktopX-ray Diffractometer) equipped with monochromated Cu Ka (λ = 0.15406 nm) radiation at a scanning speed of 8°·min^−1^ in the 2θ range of 5°–80° at room temperature. Transmissionelectron microscopy (TEM) and high-resolution transmissionelectron microscopy (HRTEM) were carried out on a JEOL Model JEM-2100-F microscope at an accelerating voltage of 200 kV. Nitrogen adsorption/desorption isotherms were measured at −196°C on a TriStar II 3020 volumetric adsorption analyzer. Prior to the measurement, the obtained samples were degassed under high vacuum at 120 °C for 5 h. The surface area was calculated from the adsorption isotherm plot in the range of relative pressure from 0.05 to 0.30 by BET method. UV-Vis adsorption spectra were analyzed by the TU-1901 spectrophotometer (Beijing Purkinje General Instrument Co. Ltd.).

### 3.3. Photocatalytic Experiments

Seventy-five milligrams of pH0.5-TiO_2_ and pH1.0-TiO_2_ were dispersed in 150 mL of methylene blue (MB, obtained from Beijing Chemical Works) solution (2.92 × 10^−5^ mol·L^−1^) under dark and stirring conditions, respectively. The resulted suspension was stirred for 2 h in the darkness to reach adsorption–desorption equilibrium. The irradiation was carried out under stirring with a 175 W ultraviolet lamp (Shanghai Mingyao Glass Hardware Tools, Shanghai, China) located 40 cm horizontally from the sample. The concentration change of MB in the suspension solution with irradiation time was determined by using a TU-1901 spectrophotometer. For the comparison, the commercially available Degussa P25-TiO_2_ sample (Nippon Aerosil Ltd., Tokyo, Japan) with an average particle size of 27.8 nm and a composition of ~87% anatase, ~13% rutile phase, and BET surface area of 52.5 cm^2^/g was used as the standard sample for the photocatalytic measurement.

## 4. Conclusions

Anatase/rutile mixed-phase TiO_2_ nanocomposites have been successfully synthesized through a simple sol-gel route with further calcinations by using titanium tetrachloride as the titanium source, analytical ethanol and deionized water as the solvent, and ammonium fluoride as the capping agent. The relative amount of anatase and rutile can be slightly changed by adjusting the pH value of the suspension. The anatase/rutile mixed-phase TiO_2_ nanocomposites contained rod-like, cuboid, and some irregularly shaped anatase nanoparticles (exposed {101} facets) with sizes range from tens to more than 100 nanometers, and rod-like rutile nanoparticles (exposed {110} facets) with sizes range from tens to more than 100 nanometers. The degradation experiments demonstrate that the mixed-phase TiO_2_ nanocomposites show better photocatalytic activity than commercial Degussa P25-TiO_2_ under UV light irradiation because of the synergistic effect of the optimum anatase to rutile ratio and the specific exposed crystal surface. The anatase/rutile TiO_2_ nanocomposites obtained at pH = 1.0 (pH1.0-TiO_2_) show the best photocatalytic activity, which can be attributed to the optimal heterojunction structure, the smaller average particle size, and presence of a specific exposed crystal surface. This work presents a simple and economical method to prepare efficient anatase/rutile TiO_2_ photocatalysts for wide applications in the removal of the organic dyes from colored wastewater.

## Figures and Tables

**Figure 1 molecules-24-02996-f001:**
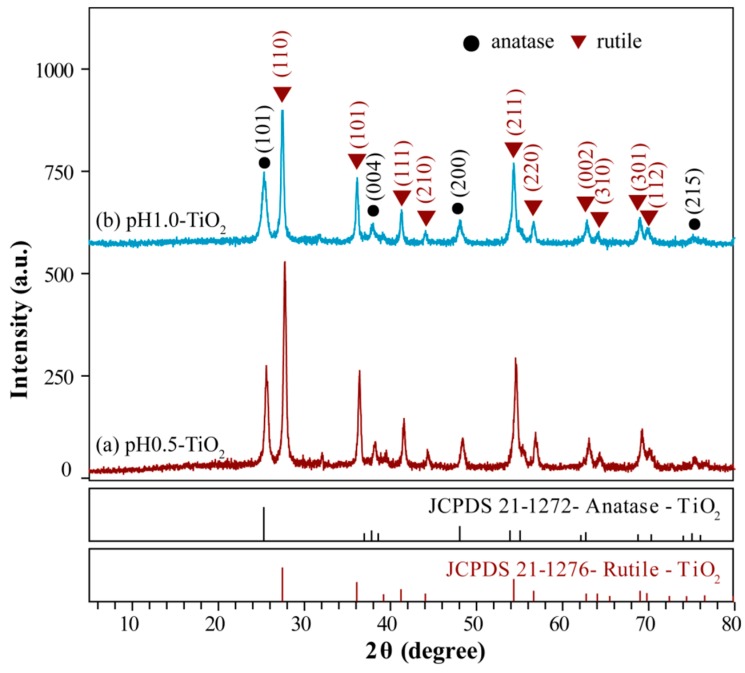
XRD patterns of (**a**) pH0.5-TiO_2_, (**b**) pH1.0-TiO_2_.

**Figure 2 molecules-24-02996-f002:**
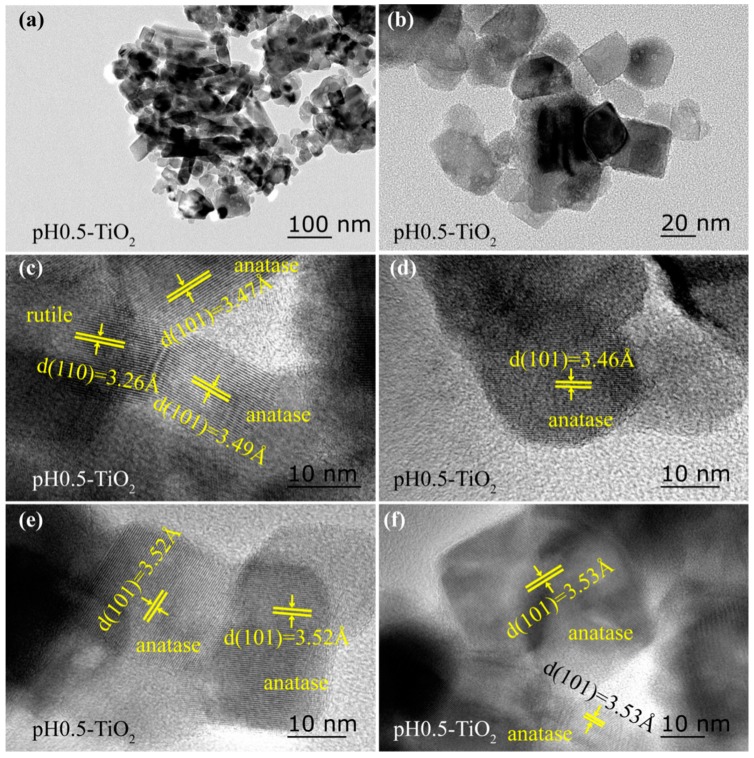
TEM (**a**,**b**) and HRTEM (**c**–**f**) images of the TiO_2_ sample obtained at pH0.5 under hydrothermal conditions.

**Figure 3 molecules-24-02996-f003:**
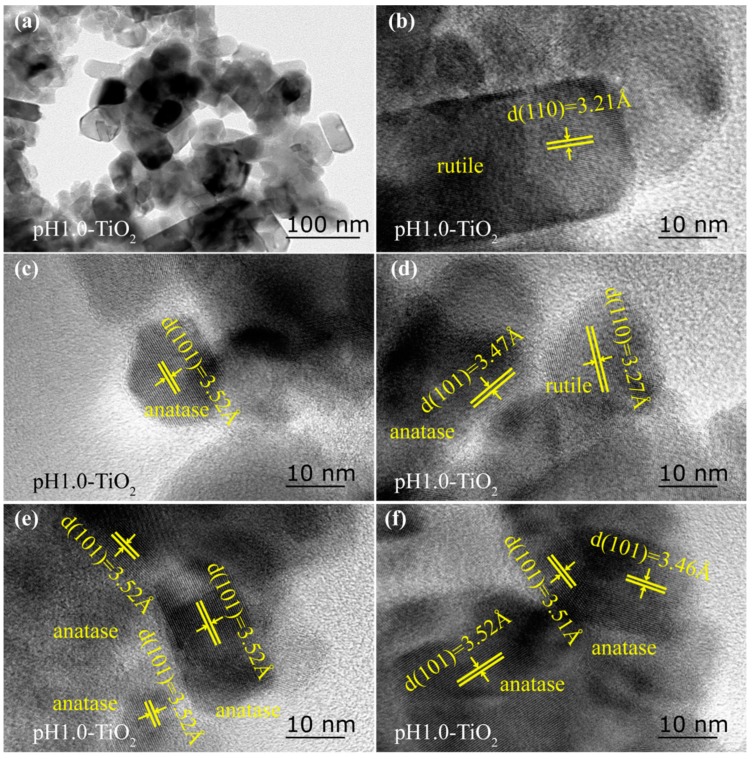
TEM (**a**) and HRTEM (**b**–**f**) images of the TiO_2_ sample obtained at pH1.0 under hydrothermal conditions.

**Figure 4 molecules-24-02996-f004:**
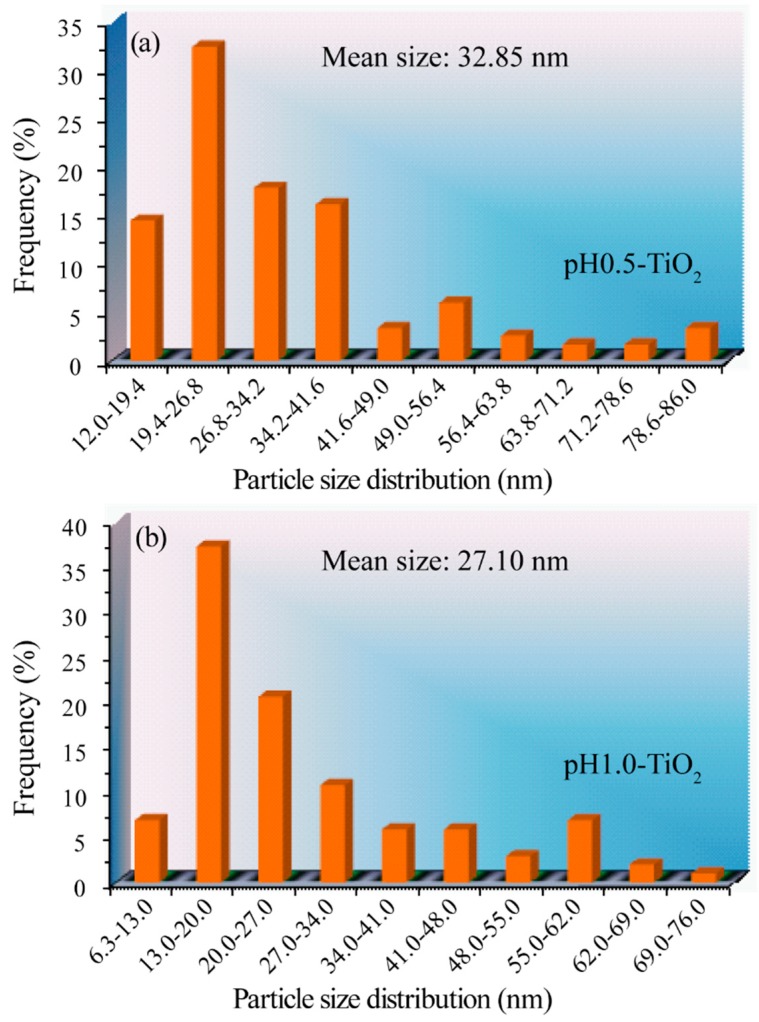
Particle size distribution of (**a**) pH0.5-TiO_2_ and (**b**) pH1.0-TiO_2_.

**Figure 5 molecules-24-02996-f005:**
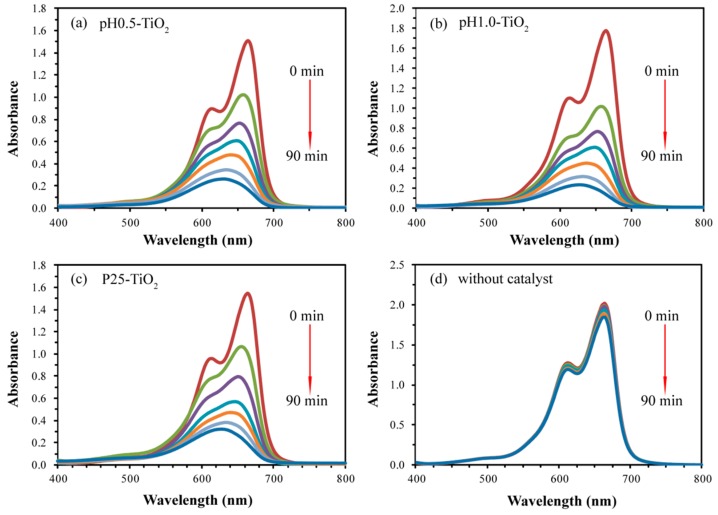
UV-vis spectral changes of MB solutions as a functional of UV irradiation time in the presence of (**a**) pH0.5-TiO_2_, (**b**) pH1.0-TiO_2_, and (**c**) P25-TiO_2_, and (**d**) absence of any photocatalysts.

**Figure 6 molecules-24-02996-f006:**
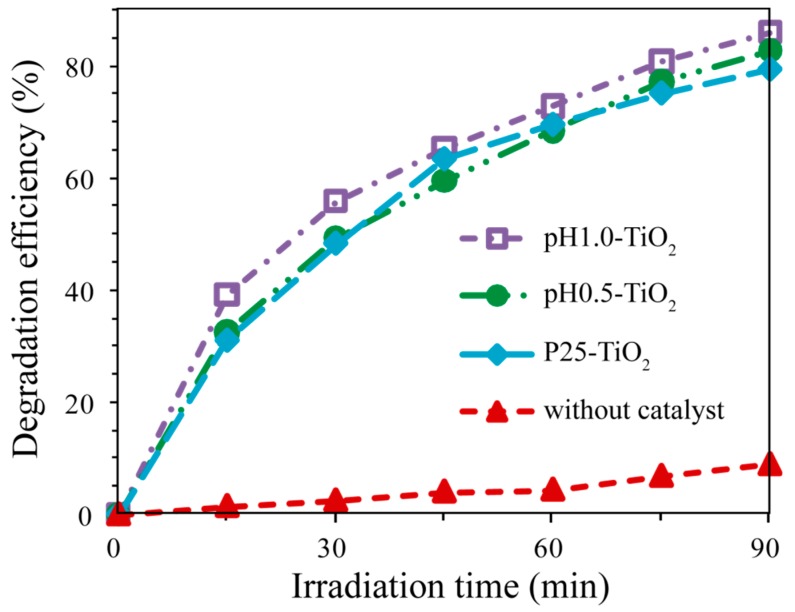
Time-dependent photodegradation profiles of MB over pH0.5-TiO_2_, pH1.0-TiO_2_, and P25-TiO_2_ photocatalysts, and a blank sample without any catalysts under UV irradiation.

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
