# Peer review of "Facile Formation of Anatase/Rutile TiO2 Nanocomposites with Enhanced Photocatalytic Activity"

_molecules, 2019, doi:10.3390/molecules24162996_

Round 1

Reviewer 1 Report

This manuscript needs significant improvements to recommend for the publication in Molecules journal

Page 1, commercial commercial

Page 2, two most comment used methods. It is commonly

Line 73, N-doped anatase TiO2 [14];

Line 77, [Ti1.73O4]1.07-

After reading the following sentence “However, the band gap of anatase is about 3.2 eV, which can only be excited by photons with wavelengths below 387 nm, i.e., anatase can show a photocatalytic activity only under UV light irradiation, while UV light accounts for only a small fraction (~5%) of the solar energy [1]”, I thought that the main objective of this study is to develop a novel titania material that is efficient under visible light irradiation. Surprisingly, the activity studies were carried out under UV light irradiation. If so, what is the novelty of this study?

Line 100, Figure 2, actually it is figure 1. Have to change it in many places of the XRD section.

Line 108, 2θ = 27.74°、36.42°、41.58°、44.40°、54.64°、56.56°、

Figure 2 must be displayed after TEM images.

The authors reported that the presence of specific exposed crystal surface on the surface of TiO2 is one of the key reasons for high efficiency. But, they did not provide the explanation i.e. how they can improve the efficiency of TiO2? Are they provide low band gap? If so, how much? Are they control recombination rate of electron-hole pairs? If so, how?

In results and discussion, each section should be improved significantly with detailed explanation citing the relevant literature articles.

I don’t see the significance of BET surface area on the catalytic efficiency of TiO2 as all the catalysts contain around 50 m2/g. the authors should keep in mind that the experimental error for BET analysis is around ±10% and sometimes more than that.  

The English usage should be improved substantially and there are many typo errors.

Reviewer 2 Report

In the literature, there are a few researches comparing P25 with newly TiO2 based materials, it would be welcomed to present a paragraph about them in the Introduction. The novelty of the research is not clearly highlighted, please update this. In the XRD spectra, there are some unidentified peaks (at around 64 and 72 2θ), where they come from? The authors often use the “high specific surface area” expression. In present case the SBET is around 50m2/g in both cases, for me it is not seems to be a high specific surface area. There are examples of 250m2/g specific surface area for crystalline TiO2 materials. I am not sure that in this case the most important influence on the photocatalytic activity is given by the 5 m2/g difference in the specific surface area. May be the crystallites type, form and size are more important.

Round 2

Reviewer 1 Report

The manuscript is improved significantly and hence, I recommend it for publication after English corrections.